# Sustainable Processing of Floral Bio-Residues of Saffron (*Crocus sativus* L.) for Valuable Biorefinery Products

**DOI:** 10.3390/plants10030523

**Published:** 2021-03-11

**Authors:** Stefania Stelluti, Matteo Caser, Sonia Demasi, Valentina Scariot

**Affiliations:** Department of Agricultural, Forest and Food Sciences, University of Torino, Largo Paolo Braccini 2, 10095 Grugliasco, Italy; stefania.stelluti@unito.it (S.S.); sonia.demasi@unito.it (S.D.); valentina.scariot@unito.it (V.S.)

**Keywords:** dried tepals, total phenolic content, total anthocyanin content, antioxidant activity, vitamin C, ultrasound assisted extraction, biorefining

## Abstract

Tepals constitute the most abundant bio-residues of saffron (*Crocus sativus* L.). As they are a natural source of polyphenols with antioxidant properties, they could be processed to generate valuable biorefinery products for applications in the pharmaceutical, cosmetic, and food industries, becoming a new source of income while reducing bio-waste. Proper storage of by-products is important in biorefining and dehydration is widely used in the herb sector, especially for highly perishable harvested flowers. This study aimed to deepen the phytochemical composition of dried saffron tepals and to investigate whether this was influenced by the extraction technique. In particular, the conventional maceration was compared with the Ultrasound Assisted Extraction (UAE), using different solvents (water and three methanol concentrations, i.e., 20%, 50%, and 80%). Compared to the spice, the dried saffron tepals showed a lower content of total phenolics (average value 1127.94 ± 32.34 mg GAE 100 g^−1^ DW) and anthocyanins (up to 413.30 ± 137.16 mg G3G 100 g^−1^ DW), but a higher antioxidant activity, which was measured through the FRAP, ABTS, and DPPH assays. The HPLC-DAD analysis detected some phenolic compounds (i.e., ferulic acid, isoquercitrin, and quercitrin) not previously found in fresh saffron tepals. Vitamin C, already discovered in the spice, was interestingly detected also in dried tepals. Regarding the extraction technique, in most cases, UAE with safer solvents (i.e., water or low percentage of methanol) showed results of phenolic compounds and vitamin C similar to maceration, allowing an improvement in extractions by halving the time. Thus, this study demonstrated that saffron tepals can be dried maintaining their quality and that green extractions can be adopted to obtain high yields of valuable antioxidant phytochemicals, meeting the requirement for a sustainable biorefining.

## 1. Introduction

Saffron (*Crocus sativus* L.), of the Iridaceae family, is a geophyte widely cultivated for its red-scarlet stigmas that, once dried, form the most expensive spice in the world ($40–50 g^−1^, [1]). Saffron spice has nutritional and phytochemical compounds such as terpenes, phenols, and vitamin C, with antioxidant [2,3] and therapeutic [4,5] properties.

Saffron, being a triploid (2*n* = 3*x* = 24) sterile species, is propagated through underground clonal corms [6]. In Mediterranean climate regions flowering occurs for few weeks from early to late autumn and is mainly controlled by seasonal thermoperiodicity, soil water content, and corm size [7,8,9]. There are one or several flowers per saffron plant, even up to 12 [6]. Flowers have a perianth of six violet tepals, three stamens, and a style culminating in three stigmas [10]. About 110 to 300 flowers are needed to obtain 1g of spice [11,12] and tepals constitute the most abundant bio-residue (~80% of total flower mass [13]).

In Persian traditional medicine, saffron tepals are considered antispasmodic, stomachic, antitumor, antidepressant, and curative of anxiety [14]. They contain proteins, fibers, fats, ashes, minerals, polyphenols, and present high antioxidant activity [14,15]. To our knowledge, vitamin C has never been detected in saffron tepals so far.

Phenolic compounds are a class of secondary metabolites commonly found in plants in which play various biological activities, such as defense against biotic and abiotic stresses, UV filters, attraction for pollinators, fruit dispersion, plant growth, and allelopathy [16,17,18,19,20]. These molecules are the main source of plant antioxidants that, by quenching ROS and reactive nitrogen species (RNS) [21], control oxidative stress, which is associated with several age-related diseases such as cancers, inflammations, and neurodegenerative disorders. This class include phenolic acids and flavonoids [22]. 

Phenolic acids include hydroxybenzoic and hydroxycinnamic acids and are the main phenolic compounds produced by plants [17]. They can enhance the organoleptic, nutritional, and antioxidant properties of food and are ancestors for the bioactive compounds used in pharmaceutic-cosmetic and food industries [17]. 

Flavonoids comprise flavonols, flavan-3-ols (such as catechins), and anthocyanins, which are the water-soluble pigments responsible for the pink-orange, red, and blue color range of flowers and fruits [22,23]. Flavonoids have antioxidant, antifungal, antibacterial, and antiviral properties and are used in the agricultural, food, and pharmaceutical-cosmetic industries [23]. In particular, anthocyanins are authorized natural food colorants in Europe with the code E-163 regardless the plant source, being considered a group of harmless substances [24].

Vitamin C is ubiquitous in plants [25]. It regulates several physiological functions, such as photosynthesis, seed germination, floral inductions, and senescence. Ascorbate is the reduced and physiologically active form, while dehydro-ascorbate is the oxidized form. The regenerative nature makes ascorbate a powerful antioxidant molecule, involved in the enzymatic and non-enzymatic defense system against oxidative stress [26]. Vitamin C is an essential dietary element since it cannot be synthetized by humans and a severe lack of its intake leads to scurvy disease [25].

The selection of suitable solvents and techniques is a basic factor to get a high extraction yield of phytochemicals from plant material [27]. Water, organic solvents, and their combination are often used to extract phenolics and vitamin C [2,3,27,28,29]. Methanol is commonly chosen for phenolic compounds due to its similar polarity, small dimension, and low density [30]. In a previous work, Caser et al. [15] compared conventional maceration with ultrasound assisted extraction (UAE) on fresh saffron tepals using water and different concentrations of methanol (20%, 50%, and 80%) as solvents. Because harvested flowers are very perishable, dehydration is a common practice to storage and preserve their quality since it inhibits enzymatic activity and limits microbial contamination [31].

This study aimed to investigate the phytochemical composition and antioxidant activity of dried saffron tepals and select the more effective and sustainable extractions. Conventional maceration was compared to the modern UAE technique that, allowing the use of safer solvents and shorter time, can reduce the energy cost of the extraction process [31,32]. A comparison of the results with a previous work performed on fresh saffron bio-residues [15] was also possible. According to the Directive 2008/122/EC, avoid waste production and use it as a resource is encouraged in the food waste management and an eco-sustainable processing is required in biorefining. As dried saffron tepals can have food, pharmaceutical, and cosmetic applications [13], investing in this by-product may be a promising approach to minimize losses [13] and potentially increase further the economic value of this crop [32].

## 2. Results

### 2.1. Qualitative and Antioxidant Properties of Dried Tepals Extracts

Total phenolic content (TPC), total anthocyanin content (TAC), and antioxidant activity by means of three assays, i.e., FRAP, ABTS, and DPPH, were evaluated for all phytoextracts (Table 1). Statistical comparisons between solvents for both extraction techniques separately were provided in Appendix A.

TAC and FRAP were significantly affected by the extractions. TPC showed an average value of 1127.94 ± 32.34 mg GAE 100 g^−1^ DW. TAC varied significantly among the different extractions, ranging from 178.39 ± 34.03 mg G3G 100 g^−1^ DW using UAE with 20% methanol to 413.30 ± 137.16 mg G3G 100 g^−1^ DW using UAE with water, with an average value of 282.90 ± 71.82 mg G3G 100 g^−1^ DW. 

The assays revealed that all extracts exhibited antioxidant activity. FRAP showed significant differences ranging from 460.05 ± 35.55 mmol Fe^2+^ Kg^−1^ DW for UAE with 20% methanol to 571.54 ± 3.21 mmol Fe^2+^ Kg^−1^ DW for maceration with water. ABTS and DPPH had an average value of 14.07 ± 0.73 μmol TE g^−1^ DW and 21.32 ± 3.45 μmol TE g^−1^ DW, respectively. 

### 2.2. Correlation Analysis

The correlation between TPC, TAC, and the antioxidant activity of the phytoextracts assessed with the FRAP, ABTS, and DPPH assays were statistically evaluated (Table 2). 

Significant correlations were only found in the case of UAE, between TPC and both ABTS (−0.69) and DPPH (−0.45), and between TAC and FRAP (0.86).

### 2.3. Bioactive Compounds from HPLC Analysis

The bioactive compounds extracted were as follows: ferulic acid (cinnamic acid); ellagic acid (benzoic acid); hyperoside, isoquercitrin, quercitrin, and rutin (flavonols); epicatechin (catechin); and vitamin C (Table 3).

Overall, the yield of phytochemicals was significantly affected by the different extractions (*p* < 0.05). 

As regards ferulic acid, 9.65 ± 2.62 mg 100 g^−1^ DW was extracted through maceration with methanol at 50%, showing no significant differences to 20%. It was not obtained by the other extractions. 

The yield of ellagic acid ranged from 1.32 ± 0.33 mg 100 g^−1^ DW for maceration with 80% methanol to 28.39 ± 4.32 mg 100 g^−1^ DW for UAE with 50% methanol, which was not significantly different to maceration with water and UAE with both 20% and 80% methanol. No extraction was performed by maceration with 50% methanol.

Hyperoside was obtained by all extractions. Its values varied from 4.35 ± 1.04 mg 100 g^−1^ DW regarding maceration with water to 27.26 ± 4.29 mg 100 g^−1^ DW for UAE with 50% methanol, which was not significantly different to the other methanol concentrations and maceration with both 20% and 80% methanol. 

Isoquercitrin ranged from 0.22 ± 0.12 mg 100 g^−1^ DW for maceration with 20% methanol to 7.82 ± 3.09 mg 100 g^−1^ DW as regards maceration with 80% methanol, that not significantly differed to maceration with 80% methanol and UAE with both 20% and 50% methanol. It was not extracted by UAE with both water and 80% methanol.

Quercitrin was only achieved by means of maceration with all methanol levels and UAE with 50% methanol. Maceration with 50% methanol gave 9.27 ± 3.47 mg 100 g^−1^ DW, without significant differences with the other extractions.

The yield of rutin varied from 0.32 ± 0.31 mg 100 g^−1^ DW as regards maceration with 20% methanol to 37.61 ± 2.22 mg 100 g^−1^ DW for maceration with 80% methanol, which was not significantly different to UAE with water.

The value of epicatechin resulted 16.62 ± 15.89 for UAE with 20% methanol, without significant differences to UAE with 80% methanol. It was not attained by all other extractions. 

Regarding vitamin C, 33.72 ± 0.89 mg 100 g^−1^ DW was gain through maceration with 20% methanol without significant differences to maceration with water and UAE with both water and 20% methanol. No extraction was achieved by all other extractions. 

The phytochemical profile of the extracts depended on the extractions (Table 2). For example, all the molecules except epicatechin were obtained using maceration with 20% methanol, while UAE with 20% methanol extracted all the chemicals except ferulic acid and quercitrin. However, no specific method–solvent combination performed a significant higher yield of these compounds altogether (Figure 1). 

## 3. Discussion

Conventional maceration and UAE are the most common extraction methods employed to analyze the antioxidant molecules of saffron tepals (Table 4). Maceration requires a high consumption of organic solvents and long extraction times [33]. UAE is a modern and green method based on ultrasonic waves that, by increasing the permeability of plant cells through acoustic cavitation of the wall, improve the penetration of the solvent and the release of phytocompounds. UAE can allow the use of non-toxic solvents, to work at room temperature, shorter extraction times compared to maceration [31,34,35,36], which needs the addition of heat to reduce the times [32], and a lower energy cost [32].

All the extracts of dried saffron tepals presented phenolic compounds (Table 1). As showed in Table 4, other authors reported their presence in dried and fresh tepals.

The total phenolic content (TPC) was not affected by the extractions (Table 1), meaning that UAE had similar performances to maceration in half the time. This result was consistent with the absence of a better extraction to obtain all the phytochemicals analyzed with the HPLC (Figure 1). Using extraction conditions similar to this study Caser et al. [15] found that for fresh saffron tepals 80% methanol gave higher TPC. Thus, dehydrated tepals may allow the use of more ecological solvents for the extraction of phenolic compounds. UAE with water could be selected to obtain non-toxic phytoextracts to be added to foods [13]. 

The technique and the solvent used can influence the extraction performance, but other factors can also affect the concentration of phytocompounds for saffron, such as provenance [47], conditions of growth [2,3], and storage [48]. Compared to the average value of TPC (1127.94 ± 32.34 mg GAE 100 g^−1^ DW), Lahmass et al. [39] attained higher TPC from saffron dried tepals using maceration with both pure methanol (~9400 mg GAE 100 g^−1^ DW) and ethyl acetate (~4200 mg GAE 100 g^−1^ DW), while a lower TPC (~342 mg GAE 100 g^−1^ DW) was found by Amir et al. [49] using UAE with pure methanol. The values of TPC were lower compared to those found in saffron spice (1819.5 mg GAE 100 g^−1^ DW) by Caser et al. [2], but higher than those obtained from some traditional Chinese medicinal plants [50], such as *Lycopus lucidus* (~930 mg GAE 100 g^−1^ DW for methanolic extract of aerial parts), *Smilax glabra* (~740 mg GAE 100 g^−1^ DW for methanolic extract of root), *Plantago asiatica* (~850 mg GAE 100 g^−1^ DW for aqueous extract of seeds), *Lobelia chinensis* (~600 mg GAE 100 g^−1^ DW for aqueous extract of whole plant), *Lithospermum erythrorhizon* (~920 mg GAE 100 g^−1^ DW for root extract), *Dianthus superbus* (~690 mg GAE 100 g^−1^ DW for aqueous extract of aerial parts), *Curcuma longa* (~800 mg GAE 100 g^−1^ DW for rhizome methanolic extract), and *Zingiber officinale* (~740 mg GAE 100 g^−1^ DW for rhizome methanolic extract), as well as from some common vegetables and fruits [50], such as spinach (*Spinacia oleracea*, ~900 mg GAE 100 g^−1^ DW for methanolic extract) and lettuce (*Lactuca sativa*, ~780 mg GAE 100 g^−1^ DW for methanolic extract). 

The total anthocyanins content (TAC) had an average value of 282.90 ± 71.82 mg G3G 100 g^−1^ DW (Table 1). Even in the case of anthocyanins, UAE achieved a yield similar to maceration in half the extraction time. A higher result was seen for UAE with water (413.30 ± 137.16 mg G3G 100 g^−1^ DW), showing a percentage variation from the average yield of +46%. This is in contrast with the results obtained from fresh saffron tepals [15], for which 80% methanol performed better meaning that even in the case of anthocyanins dehydration may allow the use of greener solvents. The values of TAC were lower than those obtained from saffron tepals by Lotfi et al. [43], which get 507.5 mg G3G 100 g^−1^ DW using maceration with acidified ethanol. TAC was also lower than that found in saffron spice (1867.3 mg G3G 100 g^−1^ DW) by Caser et al. [2], but greater compared to the contents of some highly pigmented vegetables [51], such as purple cauliflower (*Brassica oleracea* var. botrytis, 201 mg G3G 100g^−1^ DW) and red cabbage (*Brassica oleracea* var. capitata f. rubra, 199 mg G3G 100 g^−1^ DW), achieved through maceration with 0.1% HCl (*v*/*v*) in 80% methanol. Grape (*Vitis vinifera*) is one of the most common sources of these molecules [52] having a TAC ranging from 75.64 to 414.95 mg G3G 100 g^−1^ DW depending on the cultivars [53], which is comparable in its maximum values to that of our study. Since plant waste are usually used for the commercial preparation of these pigments [54], saffron tepals in both dried and fresh [15] forms could be considered a good candidate for their extraction. 

All assays evaluated, namely, FRAP, ABTS, and DPPH, revealed the presence of antioxidant activity in the extracts with average values of 520.39 ± 34.48 mmol Fe^2+^ Kg^−1^ DW for FRAP, 14.07 ± 0.73 μmol TE g^−1^ DW for ABTS, and 21.32 ± 3.45 μmol TE g^−1^ DW for DPPH (Table 1). Maceration with water obtained higher antioxidant activity measured with the FRAP method, which was not significantly different to UAE with water. This was in tune with that found by Caser et al. [2] for fresh saffron tepals. Interestingly, the antioxidant activity was higher compared to that found in saffron spice by Caser et al. [2] (338.2 mmol Fe^2+^ Kg^−1^ DW for FRAP and 4.6 μmol TE g^−1^ DW for ABTS) and Caser et al. [15] (4.64 ± 0.50 μmol TE g^−1^ DW for ABTS). The ABTS values were greater than those achieved from some traditional Chinese medicinal plants [50], such as *Chrysanthemum indicum* (3.03 μmol TE g^−1^ DW for methanolic extract of inflorescence), *Artemisia annua* (6.29 μmol TE g^−1^ DW for aqueous extract of aerial parts), *Campsis radicans* (4.462 μmol TE g^−1^ DW for methanolic extract of flower), and also compared to extracts obtained from some common vegetables and fruits [50], such as wild cabbage (*Brassica oleracea*, 1.01 μmol TE g^−1^ DW) and tomato (*Solanum lycopersicum*, 1.49 μmol TE g^−1^ DW). The DPPH values were similar to those found in mengkudu (*Morinda citrifolia*, up to ~23 μmol TE g^−1^ DW) by Thoo et al. [55].

The correlations among TPC, TAC, and the antioxidant activity were assessed (Table 2). Significant correlations were only found in the case of UAE. Even though the FRAP, ABTS, and DPPH assays have been extensively applied, they do have some limitations which can cause an imprecise estimation [56]. Since the antioxidant activity of plant extracts mainly depends on the content of phenolic compounds, it was expected a strong positive correlation among TPC and the assays [50,56,57,58]. However, TPC was either not correlated (FRAP) or negatively correlated (−0.69 ABTS and −0.45 DPPH) with the antioxidant capacity. No significant correlations among TPC and antioxidant activity was already found in extracts [59]. The negative correlation, already reported by other authors [59,60], suggests they might have opposite behavior when subjected to different solvents. The antioxidant capacity of samples may be influenced by extraction technique, solvent, pH, and metal ions [60]. Furthermore, synergistic and antagonistic interactions among the antioxidants in the extracts may interfere with the correlation, or there can be some non-phenolic molecules which can react with the Folin–Ciocalteu reagent without being free radical scavengers [59].

TAC was positively correlated with FRAP (0.86) (Table 2), indicating that anthocyanins contributed more to antioxidant capacity than TPC.

Positive correlations among the antioxidant assays were assumed [56,58], since they are all based on the same principle of electron transfer [56]. However, other authors reported the absence of significant correlation [55,59]. This might be explained by the fact that plants extracts can contain antioxidants which may react in different ways [59]. 

The heterogeneity of the yields achieved for each molecule (Table 3) would explain the absence of a single better extraction for these compounds altogether (Figure 1). Overall, except for ferulic acid and quercitrin, UAE get results higher (for ellagic acid and epicatechin) or comparable (for vitamin C) to maceration using greener solvents, i.e., water (for hyperoside and rutin) or lower methanol concentrations (20% methanol for isoquercitrin) in half the extraction time (Table 3). A unique ideal extraction does not exist, but the choice should fall on that most environmentally and economically convenient.

The bioactive compounds here identified and quantified (Table 3) deepen the knowledge on the molecular composition of dried saffron tepals. The extracts showed the same molecules (ellagic acid, hyperoside, rutin, and epicatechin) as those found in fresh tepals by Caser et al. [15] using an analysis protocol similar to this study. In addition, in the dried tepals were also detected ferulic acid, isoquercitrin, quercitrin, and vitamin C. In Caser et al. [15] the absence in fresh tepals of the chemicals here obtained might be attributed to the drying procedure [61,62], which can facilitate extractions by disrupting cell walls and causing the formation of cavities and intercellular spaces [63]. Vitamin C and rutin (yield up to 37.61 ± 2.22 mg 100 g^−1^ DW) showed the highest yields among all the compounds analyzed with HPLC. Vitamin C (up to 33.72 ± 0.89 mg 100 g^−1^ DW) was lower than in the spice (73.2 mg 100 g^−1^; Caser et al. [15]), but higher or comparable than in some freeze-dried vegetables and fruits [64], such as cabbage (*Brassica oleracea* var. capitata, up to 28.3 mg 100 g^−1^), tomato (*Solanum lycopersicum*, up to 10.1 mg 100 g^−1^), apple Fuji (*Malus domestica* ‘Fuji’, up to 3.5 mg 100 g^−1^), mango (*Mangifera indica*, up to 18.6 mg 100 g^−1^), pineapple (*Ananas comosus*, 5.0 mg 100 g^−1^), and pomelo (*Citrus maxima*, up to 31.7 mg 100 g^−1^). 

## 4. Materials and Methods

### 4.1. Plant Material

Saffron tepals were provided by the company “Lo Zafferano del Monviso” located in Martiniana Po, CN (Italy—44°23′ N 7°33′ E). Saffron plants were cultivated in open field selecting corms with horizontal diameters of 2.5 to 3.5 cm, which were planted in August 2018. The flowers were harvested in October—November 2018 and immediately air-dried. In the laboratory of the Department of Agricultural, Forest, and Food Sciences (DISAFA) of the University of Turin (Italy), the dried tepals were grinded in liquid nitrogen and stored at −80 °C until use.

### 4.2. Extraction Methods

Two extraction procedures, i.e., maceration and UAE, with deionized water or 20%, 50%, and 80% methanol in deionized water (v:v) as solvents were applied. The extractions were conducted at room temperature (ca. 21 °C) and using a powdered sample–solvent ratio of 1:50 g ml^−1^. 

As regards maceration, the samples were soaked in each solvent type and kept into a glass tube under stirring (1000 rpm) in the dark for 30 min, whereas for UAE the sample tubes were inserted into an ultrasonic extractor (Sarl Reus, Drap, France) using a frequency of 23 kHz for 15 min. The extracted solutions were all filtered with one-layer of filter paper (Whatman No. 1, Maidstone, UK) and then with a 0.45 μm PVDF syringe filter (CPS Analitica, Milano, Italy). The extracts were then stored at −20 °C for analyses. All the extractions were carried out in triplicate for each solvent and method used.

### 4.3. Spectrophotometric Analysis 

All the analysis were conducted in three replicates and performed as reported by Caser et al. [15].

#### 4.3.1. Total Phenolic Content

Total phenolic content was estimated using the Folin-Ciocalteu method. In each plastic tube 200 μL of each phytoextract were mixed with 1000 μL of diluted (1:10) Folin–Ciocalteu reagent. After 10 min of incubation at dark and room temperature, 800 μL of Na_2_CO_3_ 7.5% (*w/v*) were added to each tube. The samples were incubated at dark and room temperature for 30 min. The absorbance at 765 nm was measured by means of a UV–Vis spectrophotometer (Cary 60 UV-Vis, Agilent Technologies, Santa Clara, CA, USA). The results were expressed as mg of gallic acid equivalents (GAE) per 100 g of dry weight (mg GAE 100 g^−1^ DW).

#### 4.3.2. Total Anthocyanin Content

Total anthocyanins content was measured using the pH-differential method. Buffer solution at pH 1 (4.026 g KCl + 12.45 mL HCl 37% in a 1 L water volume) was added to 500 μL of phytoextract reaching 5 mL in each flask. The same was made in a second flask using a buffer solution at pH 4.5 (32.82 g C_2_H_3_NaO_2_ + 18 mL C_2_H_4_O_2_ in a 1 L water volume). The samples were left in the dark at room temperature for 20 min. The absorbance of both flasks was read at 515 and 700 nm at a UV–Vis spectrophotometer (Cary 60 UV-Vis, Agilent Technologies, Santa Clara, CA, USA). The total anthocyanin content was calculated using the following formula:[(*A* × *sample dilution factor* × 1000)/(*molar absorptivity* × 1)]
where *A* is [(Absorbance 515 nm − Absorbance 700 nm) at pH 1] − [(Absorbance 515 nm − Absorbance 700 nm) at pH 4.5]. The results were expressed as milligrams of cyanidin 3-O-glucoside (C3G) per 100 g of dry weight (mg C3G 100 g^−1^ DW). 

#### 4.3.3. Antioxidant Activity

The antioxidant activity was determined using the following methods:Ferric ion reducing antioxidant power (FRAP) method. The FRAP solution was produced by mixing a buffer solution at pH 3.6 (C_2_H_3_NaO_2_ + C_2_H_4_O_2_ in water), 2,4,6-tripyridyltriazine (TPTZ, 10 mM in HCl 40 mM), and FeCl_3_∙6H_2_O (20 mM). Then, 90 µL of deionized water and 900 µL of the FRAP reagent were added to 30 µL of phytoextract in each plastic tube. The samples were left at 37 °C for 30 min and the absorbance was read at 595 nm at a spectrophotometer (Cary 60 UV-Vis, Agilent Technologies, Santa Clara, CA, USA). Results were expressed as millimoles of ferrous iron equivalents per kilogram of dry weight (mmol Fe^2+^ Kg^−1^ DW).The 2,2′-azinobis (3-ethylbenzothiazoline-6-sulfonic acid) (ABTS) method. The ABTS radical cation (ABTS·) was obtained by the reaction of 7.0 mM ABTS solution with 2.45 mM K_2_S_2_O_8_ solution. The solution was incubated for 12–16 h in the dark at room temperature and then diluted with distilled water up to read an absorbance of 0.70 (±0.02) at 734 nm. 500 μL of diluted ABTS· was added to 15 µL of phytoextract and, after incubation in the dark at room temperature for 10 min, the absorbance was measured at 734 nm by means of a spectrophotometer (Cary 60 UV-Vis, Agilent Technologies, Santa Clara, CA, USA). The ABTS radical-scavenging activity was calculated as
[(*Abs0* − *Abs1*/*Abs0*) × 100]
where *Abs0* is the absorbance of the control (solution without phytoextract) and *Abs1* is the absorbance of the sample. The results were expressed as µmol of Trolox equivalents per gram of dry weight (μmol TE g^−1^ DW).The 2,2-diphenyl-1-picrylhydrazyl (DPPH) radical scavenging method. To obtain 100 μM of DPPH radical cation (DPPH·) 2 mg of DPPH were mixed up with 50 mL of MeOH, up to have an absorbance of 1.000 (±0.005) at 515 nm. Then, 1.5 mL of diluted DPPH· was added to 20 µL of phytoextract and the reaction was left in the dark at room temperature for 30 min. The absorbance was read at 515 nm at a spectrophotometer (Cary 60 UV-Vis, Agilent Technologies, Santa Clara, CA, USA). The DPPH radical-scavenging activity was calculated as
[(*Abs0* − *Abs1*/*Abs0*) × 100]
where *Abs0* is the absorbance of the control (solution without phytoextract) and *Abs1* is the absorbance of the sample. Results were expressed as μmol of Trolox equivalents per gram of dry weight (μmol TE g^−1^ DW).

### 4.4. HPLC Analysis

Qualitative and quantitative analyses of the phytoextracts were carried out using an Agilent 1200 High-Performance Liquid Chromatography coupled with an Agilent UV–Vis diode array detector (Agilent Technologies, Santa Clara, CA, USA). The chromatographic separation was made with a Kinetex C18 column (4.6 × 150 mm^2^, 5 µm, Phenomenex, Torrance, CA, USA) using several mobile phases and recording UV spectra at different wavelengths [65], as described in Table 5.

Each compound was determined comparing the retention times and UV spectra with the standards under the same chromatographic conditions as reported by Donno et al. [57]. The standards, purchased from Sigma-Aldrich (Saint Louis, MO, USA), were the following: flavonols (hyperoside, isoquercitrin, quercetin, quercitrin, and rutin), catechins (catechin and epicatechin), benzoic acids (ellagic and gallic acids), cinnamic acids (caffeic, chlorogenic, coumaric, and ferulic acids), and vitamin C (ascorbic and dehydroascorbic acids). All the analyses were performed in three replicates.

### 4.5. Statistical Analysis

All data were log transformed before the statistical analysis.

As regards the HPLC-DAD analysis, the limit of quantitation (LOQ) [57] was added when the treatments had data for only one or two replicates out of three, while the value 0 was added for the treatments without data. 

The data were analyzed with the R-studio software to identify the statistically supported differences between the different extractions. Significant mean differences were verified with one-way ANOVA (*p* < 0.05) and Tukey’s post-hoc test after checking the data for normality and homoscedasticity through Shapiro–Wilk’s test (*p* > 0.05) and Levene’s test (*p* < 0.05), respectively. The compounds non respecting the ANOVA assumptions were analyzed with Kruskal–Wallis non-parametric test (*p* < 0.05) and the post-hoc Dunn’s comparison test.

As regards the data resulting from the spectrophotometric analysis, statistical comparisons were performed using ANOVA for TPC, TAC, and FRAP and Kruskal–Wallis test for DPPH and ABTS (*p* > 0.05 in Levene’s test). Regarding the data resulting from the HPLC analysis, ANOVA was made for ellagic acid, hyperoside, and rutin, while Kruskal–Wallis test for the other compounds (*p* < 0.05 in Shapiro–Wilk’s test).

A correlation analysis between the content of total phenolics (TPC) and anthocyanins (TAC), and the assays FRAP, ABTS, and DPPH used to measure the antioxidant activity of the extracts was made with the R-studio software. The Pearson correlation was adopted when the variables were normally distributed according to the Shapiro–Wilk’s test (*p* > 0.05), otherwise the non-parametric Kendall correlation.

## 5. Conclusions

Processing food wastes and by-products to generate high-value products for industrial application has been attractive great economic and scientific interest [66,67]. The sustainable processing of biomass, including that resulting from agriculture, into valuable and safe biobased products falls within the definition of biorefining [68]. The use of renewable resources can help farmers to expand their sources of income and create new job opportunities while reducing bio-waste and the environmental footprint in a circular bioeconomy perspective [69]. 

The economic use of a by-product is related to its intrinsic, nutritional, or utility value. The dried saffron tepals are largely produced after stigma are separated from the flowers and are a natural source of antioxidant compounds. This study demonstrated that the phytochemical composition and the antioxidant capacity of extracts of dried saffron tepals is influenced by the extraction technique. In most cases, UAE with water or low percentage of methanol showed results of phenolic compounds and vitamin C similar to maceration, allowing an improvement in extractions by halving the time. Compared to the spice, the dried saffron tepals showed a lower content of total phenolics and anthocyanins, but a higher antioxidant activity. 

To produce waste-derived products, bio-waste shall either be separated and recycled at source or collected separately before undergoing treatments, which gives storage an important role (Directive 2008/122/EC). The cost of biomass and its storage are some essential factors for the economic viability of biorefineries [67]. The drying technique has always been widely used in the herbs sector to stabilize and preserve plant material, and in particular flowers, which have high perishability due to the elevated moisture content [31]. The drying treatment can preserve tepals until their possible use in food or pharmacological-cosmetic industries, for example in the preparation of soaps and cosmetic products where they can act as antioxidants or colorants in place of synthetic excipients, meeting the demand of the cosmetic sector to use products of natural origin [70].

This study allowed to compare the results with those previously attained for fresh tepals by Caser et al. [15], which performed a similar extraction protocol. This work has highlighted that, after drying, saffron tepals retain bioactive phenolics. The dehydration procedure may improve extractions as safer solvents gave higher yield of phenolic compounds. Furthermore, the HPLC-DAD analysis detected some phenolic compounds (i.e., ferulic acid, isoquercitrin, and quercitrin) not previously found in fresh tepals. Vitamin C, already discovered in the spice, was interestingly detected also in dried tepals. 

Taken together, these results assess that floral bio-residues of saffron can be sustainably processed to obtain high yields of valuable phytochemicals with potential applications in the pharmaceutical, cosmetic, and food industries, meeting the requirement for a sustainable biorefining and opening new possibilities for this by-product to become an important income source. 

## Figures and Tables

**Figure 1 plants-10-00523-f001:**
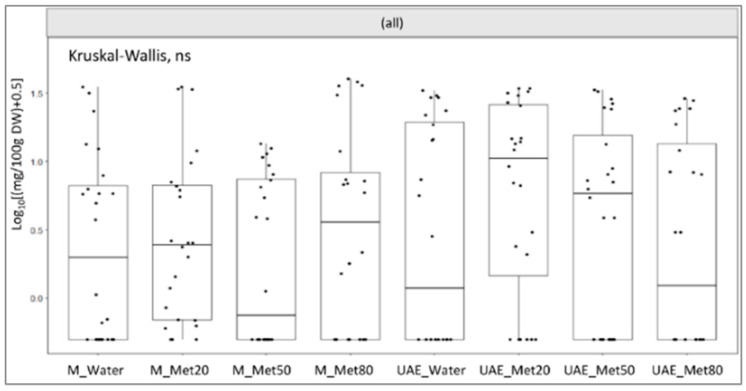
Effects of the extraction methods maceration (M) and Ultrasound Assisted Extraction (UAE) with the solvents water (M Water; UAE Water) and three concentrations of methanol in water (20%, Met20; 50%, Met50; 80% Met80; v:v) on the yield of all the compounds extracted and analyzed with HPLC-DAD. The extraction yield (mg 100 g^−1^ of dried weight, DW) is expressed as log transformation. Statistical comparisons were performed using the Kruskal–Wallis test; ns = not significant.

**Table 1 plants-10-00523-t001:** Total phenolic content (TPC), total anthocyanin content (TAC), and antioxidant activity measured with the FRAP, ABTS, and DPPH assays, in dried tepal extracts obtained through maceration (M) and Ultrasound Assisted Extraction (UAE) techniques, and the solvents water or methanol at three concentrations (20%—Met20, 50%—Met50, and 80%—Met80).

Extraction		TPC (mgGAE 100 g^−1^ DW)	TAC (mgG3G 100 g^−1^ DW)	FRAP (mmolFe^2+^ Kg^−1^ DW)	ABTS (μmolTE g^−1^ DW)	DPPH (μmolTE g^−1^ DW)
M	Water	1142.27 ± 43.52	345.04 ± 132.47 a,b	571.54 ± 3.21 a	13.82 ± 0.72	15.56 ± 2.29
M	Met20	1123.53 ± 59.86	268.13 ± 26.76 a,b	506.73 ± 13.85 b,c	14.20 ± 0.60	17.83 ± 2.46
M	Met50	1106.45 ± 9.17	300.39 ± 15.02 a,b	535.83 ± 10.30 a,b	14.62 ± 0.29	24.52 ± 2.55
M	Met80	1166.96 ± 33.15	249.13 ± 11.97 a,b	511.72 ± 22.49 a,b,c	14.29 ± 0.32	24.17 ± 1.53
UAE	Water	1150.63 ± 11.23	413.30 ± 137.16 a	556.90 ± 11.91 a,b	12.76 ± 0.81	23.55 ± 3.60
UAE	Met20	1113.27 ± 46.11	178.39 ± 34.03 b	460.05 ± 35.55 c	13.39 ± 1.46	19.35 ± 4.83
UAE	Met50	1066.89 ± 26.36	277.09 ± 49.06 a,b	506.68 ± 21.80 b,c	15.10 ± 0.38	24.58 ± 1.46
UAE	Met80	1153.49 ± 22.74	231.70 ± 30.19 a,b	513.67 ± 21.12 a,b,c	14.34 ± 0.81	21.03 ± 1.81
*p*		ns	0.01413 *	0.0002608 ***	ns	ns

Values of mean and standard deviation are reported for each variable. Statistical comparisons were performed using ANOVA for TPC, TAC, and FRAP or the non-parametric Kruskal–Wallis test for DPPH and ABTS (*p* > 0.05 in Levene’s test). Values with the same letter are not statistically different at *p* < 0.05, according to Tukey’s post-hoc test or Dunn’s post-hoc test; * *p* < 0.05; *** *p* < 0.001; ns = not significant.

**Table 2 plants-10-00523-t002:** Correlation analysis between total phenolic content (TPC), total anthocyanin content (TAC), and antioxidant activity measured with the assays FRAP, ABTS, and DPPH, as regards maceration (M) and UAE. Pearson or Kendall correlation coefficients are reported for *p*-values < 0.05.

	TPC	TAC	FRAP	ABTS
M				
FRAP	ns	ns	/	
ABTS	ns	ns	ns	/
DPPH	ns	ns	ns	ns
UAE				
FRAP	ns	0.86 ***	/	ns
ABTS	−0.69 *	ns	ns	/
DPPH	−0.45 *	ns	ns	ns

*p*-values < 0.05 show statistically significant correlations (* *p* < 0.05; *** *p* < 0.001; ns = not significant).

**Table 3 plants-10-00523-t003:** Extraction yield (mg 100 g^−1^ DW) of the compounds obtained from dried saffron tepals expressed as mg 100 g^−1^ of dried weight (DW), using the maceration (M) and Ultrasound Assisted Extraction (UAE) techniques and the solvents water or methanol at three concentrations (20%, Met20; 50%, Met50; 80%, Met80; v:v). Quantifications were obtained through the HPLC-DAD analysis.

Extractions		Cinnamic Acids	Benzoic Acids	Flavonols				Catechins	Vitamin C
		Ferulic Acid	Ellagic Acid	Hyperoside	Isoquercitrin	Quercitrin	Rutin	Epicatechin	
M	Water	0.00 ± 0.00 b	7.67 ± 3.69 a,b,c	4.35 ± 1.04 c	0.31 ± 0.22 c,d	0.00 ± 0.00 b	8.52 ± 3.91 c	0.00 ± 0.00 b	29.61 ± 6.05 a
M	Met20	1.83 ± 0.31 a	4.43 ± 4.15 c,d	5.61 ± 0.52 a,b,c	0.22 ± 0.12 c,d	6.33 ± 5.27 a	0.32 ± 0.31 d	0.00 ± 0.00 b	33.72 ± 0.89 a
M	Met50	9.65 ± 2.62 a	0.00 ± 0.00 e	5.85 ± 4.31 b,c	4.36 ± 3.49 a,b,c	9.27 ± 3.47 a	0.00 ± 0.00 d	0.00 ± 0.00 b	0.00 ± 0.00 b
M	Met80	0.00 ± 0.00 b	1.32 ± 0.33 d,e	23.93 ± 15.51 a,b,c	7.82 ± 3.09 a	6.53 ± 0.29 a	37.61 ± 2.22 a	0.00 ± 0.00 b	0.00 ± 0.00 b
UAE	Water	0.00 ± 0.00 b	8.53 ± 8.45 b,c,d	11.58 ± 4.09 a,b,c	0.00 ± 0.00 d	0.00 ± 0.00 b	28.24 ± 4.83 a,b	0.00 ± 0.00 b	26.68 ± 4.71 a
UAE	Met20	0.00 ± 0.00 b	26.74 ± 10.80 a,b	9.68 ± 6.77 a,b,c	6.46 ± 5.03 a,b	0.00 ± 0.00 b	13.46 ± 10.25 b,c	16.62 ± 15.89 a	29.17 ± 2.31 a
UAE	Met50	0.00 ± 0.00 b	28.39 ± 4.32 a	27.26 ± 4.29 a	5.57 ± 1.90 a,b,c	7.07 ± 5.12 a	7.24 ± 1.35 c	0.00 ± 0.00 b	0.00 ± 0.00 b
UAE	Met80	0.00 ± 0.00 b	23.51 ± 5.11 a,b	24.77 ± 2.25 a,b	0.00 ± 0.00 d	0.00 ± 0.00 b	9.10 ± 2.17 c	4.22 ± 2.90 a	0.00 ± 0.00 b
*p*		0.001802 **	2.235 × 10^−7^ ***	0.004662 **	0.005466 **	0.005407 **	1.452 × 10^−10^ ***	0.001995 **	0.003143 **

Values of mean ± standard deviation are reported. Statistical comparisons were performed using ANOVA (for ellagic acid, hyperoside, and rutin) or Kruskal–Wallis test (for the other compounds, *p* < 0.05 in Shapiro–Wilk’s test). Letters indicate statistical differences between the different extractions for each extracted compound. Values with the same letter are not statistically different at *p* < 0.05, according to Tukey’s or Dunn’s post-hoc test. ** *p* < 0.01; *** *p* < 0.001; ns = not significant.

**Table 4 plants-10-00523-t004:** Some representative studies of extraction methods and solvents to extract bioactive compounds from saffron tepals in different states.

Tepals State	Extraction Methods	Solvents	Bioactive Compounds	References
Fresh	Maceration, UAE	Water and methanol at 20%, 50% and 80%	Phenolic content (ellagic acid; hyperoside; rutin; epicatechin); Anthocyanin content	[15]
Dried	Soxhlet extraction	Hexane; dichloromethane; ethanol	Phenolic and Flavonoid content	[37]
Dried	Maceration	Ethanol 25%, 50%, 75%	Anthocyanin content	[38]
Dried	Maceration	Ethyl acetate; methanol	Phenolic and Flavonoid content	[39]
Dried	Maceration	Ethanol 70%	Phenolic content (kaempferol 3-O-sophoroside-7-O-glycoside, quercetin 3,4-di-O-glycoside, kaempferol di-glycoside, kaempferol 3-O-glycoside)	[40]
Dried	Maceration	Methanol	Phenolic and Flavonoid content	[41]
Dried	Maceration	Acidified (HCl) ethanol; sulfur water solution	Phenolic and Anthocyanin content (cyanidin 3,5-diglucosides; pelargonidin 3- and 5-glucosides; delphinidin di-glucosides; pelargonidin 3-glucosides; petunidin)	[42]
Dried	Maceration, Enzyme-assisted extraction	Acidified (HCl) ethanol; Enzymatic water (Pectinex)	Antocyanin content (cyanidin 3,5-diglycosides; pelargonidin 3- and 3,5-glycosides; delphinidin 3-glycosides; petunidin)	[43]
Freeze-dried	Maceration, UAE	Acidified (HCl) methanol; Acidified (HCl) ethanol 70%; Methanol 98% in formic acid	Phenolic, Flavonoid, Anthocyanin, and Crocetin esters content	[44]
Dried	Maceration, UAE	Acidified (HCl) deuterated methanol; Trifluoroacetic acid in acetonitrile 50%	Flavonoids (kaempferol di-hexoside; kaempferol 3-O-glucoside; kaempferol 3,4′-di-O-glucoside; kaempferol 3-O-β-sophoroside; kaempferol tri-hexoside; kaempferol di-hexosides; quercetin 3,4′-di-O-glucoside; isorhamnetin 3,4′-di-O-glucoside); Anthocyanins (delphinidin 3,5-di-O-β-glucoside; delphinidin 3-O-glucoside; petunidin 3,5-di-O-β-glucoside; petunidin 3-O-glucoside)	[45]
Dried	UAE	Ethanol 50%	Flavonoids (quercetin 3-O-sophoroside; kaempferol 3-O-sophoroside; kaempferol 3-O-glucoside)	[46]
Freeze-dried	Maceration, UAE	Methanol 50%; n-hexane; Methanol/KOH 20%	Flavonoid content (kaempferol 3-O-sophoroside-7-O-glucoside; kaempferol 3,7-O-diglucoside; quercetin 3,7-O-diglucoside; isorhamnetin 3,7-O-diglucoside; kaempferol 3-O-sophoroside; isorhamnetin 3-O-sophoroside; quercetin 3-O-glucoside; kaempferol 3-O-rutinoside; isorhamnetin 3-O-rutinoside; kaempferol 3-O-glucoside; kaempferol 3-O-(6??- acetyl-glycoside)-7-O-glycoside; isorhamnetin 3-O-glucoside; kaempferol 3-O-sophoroside-7-O-rhamnoside; kaempferol 3-O-(6??- acetyl-galactoside) or 3-O-(6??-acetyl- glucoside); kaempferol 3-O-(6??- acetyl-galactoside) or 3-O-(6??-acetyl- glucoside); quercetin 3-O-glucoside-7-O-rhamnoside; isorhamnetin 3-O-glucoside-7-O-rhamnoside; kaempferol 3-O-glucoside-7-O-rhamnoside); Anthocyanin content (delphinidin 3,7-O-diglucoside; petunidin 3,7-O-diglucoside; delphinidin 3-O-glucoside; petunidin 3-O-glucoside; malvidin O-glucoside); Lutein diesters	[32]
Dried	Microwave-assisted extraction	Acidified (HCl) ethanol	Anthocyanin content	[46]

**Table 5 plants-10-00523-t005:** HPLC methods and conditions.

Methods	Classes of Interest	Stationary Phase	Mobile Phase	Wavelength (nm)
A	Cinnamic acids, Flavonols	KINETEX—C18 column (4.6 × 150 mm^2^, 5 μm)	A: 10 mM KH_2_PO_4_/H_3_PO_4_, pH = 2.8 B: CH_3_CN	330
B	Benzoic acids, catechins	KINETEX—C18 column (4.6 × 150 mm^2^, 5 μm)	A: H_2_O/CH_3_OH/HCOOH (5:95:0.1 *v/v/v*), pH = 2.5 B: CH_3_OH/HCOOH (100:0.1 *v/v*)	280
C	Vitamin C	KINETEX—C18 column (4.6 × 150 mm^2^, 5 μm)	A: 5 mM C_16_H_33_N(CH_3_Br/50 mM KH_2_PO_4_, pH = 2.5 B: CH_3_OH	261, 348

Elution conditions. Method A, gradient analysis: 5% B to 21% B in 17 min + 21% B in 3 min (2 min conditioning time); flow: 1.5 mL min^−1^; Method B, gradient analysis: 3% B to 85% B in 22 min + 85% B in 1 min (2 min conditioning time); flow: 0.6 mL min^−1^; Method C, isocratic analysis: ratio of phase A and B: 95:5 in 10 min (5 min conditioning time); flow: 0.9 mL min^−1^.

## Data Availability

*Crocus sativus* L. tepals were kindly provided by the company “Lo Zafferano del Monviso” located in Martiniana Po, CN (Italy—44°23′ N 7°33′ E).

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
