# Peer review of "Sustainable Processing of Floral Bio-Residues of Saffron (Crocus sativus L.) for Valuable Biorefinery Products"

_plants, 2021, doi:10.3390/plants10030523_

Round 1
Reviewer 1 Report
In manuscript, different extraction methods were compared to optimize the bioactive compound extraction from saffron tepels. Total phenolic, anthocyanin, and antioxidant activity were measured for all extracts. The results indicate UAE have better or equivalent performance than traditional maceration for some compounds like Ellagic Acid.
Comments:
It is not surprise to see the absence of correlation between TPC and the antioxidant activity measures since there is not much variation of TPC among sample (table 1, ANOVA, not significant). But the correlations were calculated separately for Maceration and UAE, while the ANOVA were performed for all samples. Could the author perform ANOVA for Maceration and UAE separately and see if there are significant difference for samples within group? If it still no significant, I would not worry much about the absence of correlations.
In the abstract, the authors stated “In most cases, UAE demonstrated higher or similar performances to maceration in half the time”. This is conclusion can be misleading. I did not see significant difference between the UAE and Maceration in terms of total phenolics, anthocyanin, and antioxidant activity based on the post-hoc test. It can be true for some compounds such as Ellagic Acid but in most cases there is no significant difference.
Line 114 “using ANOVA or Kruskal-Wallis test. ” why rank-based non-parametric test (Kruskal-Wallis test) was used? For which variables, the Kruskal-Wallis test was used? And for which ANOVA was used?
The ‘fingerprint’ term in title can be misleading because it sounds similar to the molecular fingerprint that people used for bioactive compound prediction.
Author Response
For the preparation of the revised manuscript, we have followed all the comments and suggestions of the reviewers as stated below. We have tracked changes in the text.
English has been revised.
The abstract has been thoroughly revised.
The keywords have been modified.
We have changed the title of paragraph “4.5.” in “Spectrophotometric analysis”.
The format of the References has been improved.
Reviewer 1
REV: It is not surprised to see the absence of correlation between TPC and the antioxidant activity measures since there is not much variation of TPC among sample (table 1, ANOVA, not significant). But the correlations were calculated separately for Maceration and UAE, while the ANOVA were performed for all samples. Could the author perform ANOVA for Maceration and UAE separately and see if there are significant difference for samples within group? If it still no significant, I would not worry much about the absence of correlations.
AUT: We performed the statistical analysis for Maceration and UAE separately and did not see significant differences for samples within group.
REV: In the abstract, the authors stated “In most cases, UAE demonstrated higher or similar performances to maceration in half the time”. This is conclusion can be misleading. I did not see significant difference between the UAE and Maceration in terms of total phenolics, anthocyanin, and antioxidant activity based on the post-hoc test. It can be true for some compounds such as Ellagic Acid but in most cases there is no significant difference.
AUT: We changed the sentence in “Regarding the extraction technique, in most cases, UAE with safer solvents (i.e. water or low percentage of methanol) showed results of phenolic compounds and vitamin C similar to maceration, allowing an improvement in extractions by halving the time.”
REV: Line 114 “using ANOVA or Kruskal-Wallis test” why rank-based non-parametric test (Kruskal-Wallis test) was used? For which variables, the Kruskal-Wallis test was used? And for which ANOVA was used?
AUT: We modified the sentence in “Statistical comparisons were performed using ANOVA for TPC, TAC, and FRAP, or the non-parametric Kruskal-Wallis test for DPPH and ABTS (p > 0.05 in Levene’s test)”. We have also modified the text of Table 3 and of the paragraph “4.3. Statistical Analysis”, accordingly.
REV: The ‘fingerprint’ term in title can be misleading because it sounds similar to the molecular fingerprint that people used for bioactive compound prediction.
AUT: We changed the title in “Sustainable processing of floral bio-residues of Saffron (Crocus sativus L.) for valuable biorefinery products”.

Reviewer 2 Report
The manuscript "Optimization of Crocus sativus L. By-product: Antioxidant Compound Fingerprint and Bioactivity Evaluation of Extracts Obtained from Dried Tepals" is overall in a good shape and well-written. I have found only many language mistakes and problems with the format, and introduction that require attention to be solved.
However, my problem is related to the novelty of the study. All these compounds were known about their presence in plants, even when the extraction of leaf material is done with an organic solvent. We know that leaves have chlorophyll don't need to do MS analysis to know that.
About the title, the manuscript has no insight about the metabolome, it just a description. When I read the title I was induced to think about some things that were not found in the text, this is frustrating! I was expecting a better characterization of the molecules, their biological activity of them to justify the name of the medicinal plant, and to know how compounds are involved in that purpose.
My central questions to the authors are: What is the novelty of this work? What are the scientific contributions of this work?
Please, don´t get me wrong, my intent is to help the author to improve the manuscript.
Then, my suggestion is to make a deep characterization with chemical and biochemical experiments and also make biological activities to understand which compounds are involved in this action.

Author Response
For the preparation of the revised manuscript, we have followed all the comments and suggestions of the reviewers as stated below. We have tracked changes in the text.
English has been revised.
The abstract has been thoroughly revised.
The keywords have been modified.
We have changed the title of paragraph “4.5.” in “Spectrophotometric analysis”.
The format of the References has been improved.
Reviewer 2
REV: The manuscript "Optimization of Crocus sativus L. By-product: Antioxidant Compound Fingerprint and Bioactivity Evaluation of Extracts Obtained from Dried Tepals" is overall in a good shape and well-written. I have found only many language mistakes and problems with the format, and introduction that require attention to be solved.
AUT: We checked the entire manuscript for language and format issues and we corrected them.
REV: My central questions to the authors are: What is the novelty of this work? What are the scientific contributions of this work?
AUT: We modified the aim of the introduction, the conclusions, and the abstract to make it more clear.

Round 2
Reviewer 1 Report
"AUT: We performed the statistical analysis for Maceration and UAE separately and did not see significant differences for samples within group."
Could you provide the p-values for these analysis either in table 1 or as a supplemental table?
"ANOVA for TPC, TAC, and FRAP, or the non-parametric Kruskal-Wallis test for DPPH and ABTS"
What is the justification for using different statistic tests?
Author Response
We thank for the comments and suggestions.
Reviewer 1
REV: Could you provide the p-values for these analysis either in table 1 or as a supplemental table?
AUT: We provided a supplemental Table for the statistical analysis required. This analysis is interesting because it shows the differences in yield based on the solvents used for each extraction technique, but it does not allow to make a comparison between the extractions altogether.
We added the following sentence: “Statistical comparisons between solvents for both extraction techniques separately were provided in Supplementary Table 1.” (Line 102).
REV: What is the justification for using different statistic tests?
AUT: The non-parametric Kruskal Wallis test and the post-hoc Dunn’s comparison test (which was performed for p-value <0.05 in the Kruskal Wallis test) were used when the data did not meet the ANOVA assumptions for normality (tested with Saphiro-Wilk's test, p > 0.05) and / or homoskedasticity (verified with Levene's test, p <0.05).
We described the use of the statistical test in line 414 as follows: “Significant mean differences were verified with one-way ANOVA (p < 0.05) and Tukey’s post-hoc test after checking the data for normality and homoscedasticity through Saphiro-Wilk’s test (p > 0.05) and Levene’s test (p < 0.05), respectively. The compounds non respecting the ANOVA assumptions were analysed with Kruskal-Wallis non-parametric test (p < 0.05) and the post-hoc Dunn’s comparison test.”
We remain available to clarify any issue or answer that Reviewers or Editors may raise.
Best regards,
Sincerely yours,
Stefania Stelluti
Matteo Caser
Sonia Demasi and
Valentina Scariot
Department of Agricultural, Forest and Food Sciences
University of Turin
Largo Paolo Braccini, 2
10095, Grugliasco (TO)
Italy
Phone number: +039-011/6708935
Fax number: +039-011/6708798
e-mail: matteo.caser@unito.it

Reviewer 2 Report
Accept as it is.
Author Response
We thank for the comments and suggestions.
Reviewer 2
REV: Accept as it is.
AUT: We thank Reviewer 2.
We remain available to clarify any issue or answer that Reviewers or Editors may raise.
Best regards,
Sincerely yours,
Stefania Stelluti
Matteo Caser
Sonia Demasi and
Valentina Scariot
Department of Agricultural, Forest and Food Sciences
University of Turin
Largo Paolo Braccini, 2
10095, Grugliasco (TO)
Italy
Phone number: +039-011/6708935
Fax number: +039-011/6708798
e-mail: matteo.caser@unito.it
Round 3
Reviewer 1 Report
Thanks for the clarification.